Mitochondrial destiny in type 2 diabetes: the effects of oxidative stress on the dynamics and biogenesis of mitochondria

Skuratovskaia Daria dariask@list.ru
Komar Alexandra
Vulf Maria
Litvinova Larisa
Immanuel Kant Baltic Federal University , Kaliningrad , Russian Federation
Gould Gwyn
Electronic publication date: 2020 Aug 25
Publication date: 2020
Volume: 8
Electronic Location ID: e9741
Received 2020 Apr 13; Accepted 2020 Jul 26
Copyright: ©2020 Skuratovskaia et al.
Copyright year: 2020
Copyright holder: Skuratovskaia et al.
License: This is an open access article distributed under the terms of the Creative Commons Attribution License, which permits unrestricted use, distribution, reproduction and adaptation in any medium and for any purpose provided that it is properly attributed. For attribution, the original author(s), title, publication source (PeerJ) and either DOI or URL of the article must be cited.
License URL: https://creativecommons.org/licenses/by/4.0/

Keywords: Obesity, Insulin resistance, Mitochondrial fission and fusion, Inflammation, MOTS-C

Funding: Russian Foundation for Basic Research 18-015-00084-a Russian Foundation for Basic Research and Kaliningrad Region 19-415-393004 - r_mol_a 9-44-390005 - r_a The State Assignment 075-03-2020-080 Scientific Schools of the Russian Federation 2495.2020.7 The research was supported by Russian Foundation for Basic Research (No. 18-015-00084-a to Larisa Litvinova), by the Russian Foundation for Basic Research and Kaliningrad Region (No.19-415-393004r_mol_a and No.19-44-390005r_a to Daria Skuratovskaia), by State Assignment (No. 075-03-2020-080 to Larisa Litvinova), and by state support of leading scientific schools of the Russian Federation (No. 2495.2020.7 to Larisa Litvinova). The funders had no role in study design, data collection and analysis, decision to publish, or preparation of the manuscript.

==============================
Background

One reason for the development of insulin resistance is the chronic inflammation in obesity.

Materials & Methods

Scientific articles in the field of knowledge on the involvement of mitochondria and mitochondrial DNA (mtDNA) in obesity and type 2 diabetes were analyzed.

Results

Oxidative stress developed during obesity contributes to the formation of peroxynitrite, which causes cytochrome C-related damage in the mitochondrial electron transfer chain and increases the production of reactive oxygen species (ROS), which is associated with the development of type 2 diabetes. Oxidative stress contributes to the nuclease activity of the mitochondrial matrix, which leads to the accumulation of cleaved fragments and an increase in heteroplasmy. Mitochondrial dysfunction and mtDNA variations during insulin resistance may be connected with a change in ATP levels, generation of ROS, mitochondrial division/fusion and mitophagy. This review discusses the main role of mitochondria in the development of insulin resistance, which leads to pathological processes in insulin-dependent tissues, and considers potential therapeutic directions based on the modulation of mitochondrial biogenesis. In this regard, the development of drugs aimed at the regulation of these processes is gaining attention.

Conclusion

Changes in the mtDNA copy number can help to protect mitochondria from severe damage during conditions of increased oxidative stress. Mitochondrial proteome studies are conducted to search for potential therapeutic targets. The use of mitochondrial peptides encoded by mtDNA also represents a promising new approach to therapy.

Introduction

Diets that are high in fat and/or sucrose can lead to insulin resistance (IR) and are often associated with mitochondrial dysfunction. Both types of diet can be considered extreme and lead to a significant change in the normal balance between substrates of cellular intermediary metabolism (Jorgensen et al., 2017). As a rule, this is also associated with excess substrate in a particular diet, that is, fatty acids or sucrose (glucose plus fructose), creating an overload in the cell. Therefore, it is likely that this type of altered metabolic condition affects the normal regulatory mechanisms of cells, particularly in insulin-dependent tissues, such as adipose tissue, muscle, and liver.

Mitochondria are double-membrane organelles. Process oxidative phosphorylation (OXPHOS) enables ATP generation in mitochondria. Membrane electrochemical gradient stimulates the synthesis of OXPHOS and ATP. This gradient includes electrons are transferred to nicotinamide adenine dinucleotide (NAD+) and flavin adenine dinucleotide (FAD), and NADH and FADH2 are produced. These molecules are formed as a result of the course of the tricarboxylic acid (TCA) cycle, which is generated by the oxidation process of acetyl coenzyme A (acetyl-CoA). In turn, acetyl-CoA is formed by the react oxidation of fatty acids and pyruvate (Stanley, Recchia & Lopaschuk, 2005). Electrons are accepted and transported via components of the electron transfer chain (ETC) in the inner mitochondrial membrane. The electron transfer along the ETC is associated with proton transfer through the inner membrane. This process establishes an electrochemical gradient, which is necessary to regulate the synthesis of ATP by the ATP synthase, FoF1-ATPase (Guo et al., 2018).

Mitochondria in adipocytes can play a significant role in the regulation of energy homeostasis of the whole body, in controlling sensitivity to insulin and glucose metabolism. Mitochondria are a regulator of lipolysis in adipocytes and the main source of ATP in cells. Mitochondria are key organelles that control the physiological role of adipocytes: differentiation, lipid homeostasis, insulin sensitivity, oxidative ability, adaptive thermogenesis. Disruption of the functioning of mitochondria in insulin-dependent tissues leads to an energy crisis, which underlies the formation of IR (Lee et al., 2019).

Given the potentially important role of mitochondrial dysfunction in the pathogenesis of many diseases and the process of ageing, an understanding of the molecular mechanisms responsible for mitochondrial biogenesis and function could lead to important new therapeutic targets.

Systematization of data on this topic allows you to get an overview of the current state of research in this area as a whole, rather than detailed knowledge of the area in question, as well as to get links to the most useful primary sources

This review provides a solid starting point for all community members interested in a particular research topic and is a source of information for practitioners who seek the latest evidence that can guide them in their decision making and work.

Survey Methodology

We have provided comprehensive and objective coverage of topics by synthesizing existing literature and identifying knowledge gaps in empirical research. We used the search engines PubMed, MedLine, ClinicalTrials, Kegg data base.

We searched for keywords and word combinations: mtDNA, deletion mtDNA, mitochondria, oxidative stress, obesity, type 2 diabetes, peptide, proteome, genome, MOTS-C, TNF, IL, inflammation, glucotoxicity and lipotoxicity, biogenesis, mitophagy, dynamics of mitochondria, ROS, autophagy, mitochondria-dependent apoptosis.

We selected articles on the basis of certain eligibility criteria - the quality of the work performed, based on the ratings of journals, and this is a search depth of research articles of no more than 5 years.

Factors contributing to the development of mitochondrial dysfunction and insulin resistance

The mechanisms for the development of IR have been studied extensively; however, consensus on a single mechanism has not yet been reached. It remains unclear whether mitochondrial dysfunction is a result of IR or whether IR itself causes mitochondrial dysfunction. One generally accepted theory for the development of IR suggests that inflammation contributes to the increased production of reactive oxygen species reactive oxygen species (ROS). In obesity, adipocytes undergo hypertrophy, promoting the release of free fatty acids during lipolysis and the development of hypoxia in adipose tissue (Lee et al., 2019; Snodgrass et al., 2016). Immune cells and macrophages are then recruited to adipose tissue by inflammatory mediators (Snodgrass et al., 2016). The necrotic adipocytes are phagocytosed by macrophages, which produce chemokines and pro-inflammatory mediators (Fig. 1) (Snodgrass et al., 2016; Bergmann & Sypniewska, 2013).

Figure 1 The development of mitochondrial dysfunction and insulin resistance on the background of obesity.

(A) Adipose tissue secretes pro-inflammatory mediators and free fatty acids (FFAs) due to hypoxia, generating chronic inflammation. High levels of FFAs, DAMPS/PAMPS, leptin, TNFα, and IL-6 block insulin signalling. (B) In mitochondria, the electron transport chain (ETC) is disrupted, which is characterized by a decrease in ATP production and an increase in ROS. Oxidative stress damages DNA, RNA, proteins, and, in particular, mtDNA, which leads to the formation of deletions and a decrease in the functional activity of mitochondria. Under the action of oxidative stress, the process of mitochondrial fission and fusion is launched. These processes mix the contents of partially damaged mitochondria and contribute to quality control by enabling the removal of damaged mitochondria and creating new mitochondria. Impaired mitochondrial function in insulin-dependent tissues, in addition to contributing to the formation of inflammation, leads to an energy crisis that underlies insulin resistance. This figure has been created by modifying the templates from Servier Medical Art (Servier Medical Art, 2020); CC BY 3.0 Unported.

Transcription factors involved in the pro-inflammatory pathways, Nuclear factor kappa B (NF-kB), signal transducer and activator of transcription (STAT), and Activator protein (AP-1), are activated, triggering subsequent signalling cascades that mediate the expression of proteins, which then inhibit the insulin signalling pathway, contributing to the development of IR (Lee et al., 2019; Bergmann & Sypniewska, 2013). Signalling molecules can interact with effectors on the surface of mitochondria (Lee et al., 2019; Litvinova et al., 2015). Excessive peroxynitrite formation causes mitochondrial ETC to release cytochrome C and increases the production of ROS (Fig. 1) (Münzel et al., 2010), which are associated with the development of type 2 diabetes (Litvinova et al., 2015).

For example, the tumour necrosis factor-alfa (TNF-α) induces mitochondrial dysfunction, decreasing the activity of complex III in ETC, increasing the production of ROS, and causing damage to mitochondrial DNA (mtDNA). Mitochondrial superoxide anion is a precursor of most ROS and a mediator in the reaction of oxidative stress. Superoxide dismutase leads to the formation of hydrogen peroxide and a hydroxyl radical. There radicals causing damage to various mitochondrial and cellular components. Such damage in the mitochondria can lead to a decrease in the affinity of mitochondrial proteins for substrates or coenzymes (Litvinova et al., 2015).

There is no doubt that genetic mutations in mtDNA, which can be either innate or acquired, are associated with the development of IR; numerous studies have shown the relationship between polymorphisms and the development of metabolic disorders. For example, dysfunctional mtDNA reduces the efficiency of OXPHOS and ATP synthesis (Pagel-Langenickel et al., 2010). Under oxidative stress, mitochondria generate superoxide, hydrogen peroxide, and hydroxyl radicals, thereby contributing to protein, DNA, RNA, and lipid damage (Yuzefovych et al., 2013; Indo et al., 2015).

Oxidative stress contributes to the nuclease activity of the mitochondrial matrix, which leads to the accumulation of cleaved fragments and an increased level of heteroplasmy. Under oxidative stress, mtDNA is damaged by deletions (Mishra et al., 2016; Herbst et al., 2016). Patients with mt DNA mutations are characterized by decreased functioning of pancreatic beta cells and decreased glucose-stimulated insulin secretion (Kim et al., 2017; Jiang et al., 2017).

Ye (2013) proposed the relationship between energy metabolism in mitochondria and IR. According to this hypothesis, IR is the result of excess energy in cells (Ye, 2013), and a decrease in the functional activity of mitochondria is considered to be an adaptive mechanism for the soft separation of the ETC, which protects mitochondria from the vicious cycle of ROS production. The author suggested that this mechanism offers a promising approach to the treatment of IR (Ye, 2013).

The effects of certain drugs for the treatment of type 2 diabetes prove this hypothesis. Among the mechanisms of action of these drugs that lead to improved insulin sensitivity, is the suppression of ATP generation in mitochondria (Pagel-Langenickel et al., 2010; Ye, 2013). In addition, effective weight loss treatments, exercise, and a low-calorie diet are effective treatments for insulin-dependent ATP in insulin-sensitive tissues (Ye, 2013). The new concept assumes the existence of a unified cellular and molecular mechanism of IR in obesity.

Therefore, mitochondria play a key role in the pathogenesis of IR by regulating lipolysis in adipocytes and serving as the main sources of ATP in cells (Fig. 1) (Serra et al., 2013). Impaired mitochondrial function in insulin-dependent tissues leads to an energy crisis that underlies the development of IR (Pagel-Langenickel et al., 2010; Maechler & Wollheim, 2001).

Disturbances of mitochondrial fission and fusion during oxidative stress

Mitochondrial fission and fusion play a crucial role in maintaining functional mitochondria when cells experience metabolic or environmental stress. Fusion helps to alleviate stress by combining the contents of partially damaged mitochondria, whereas fission is necessary to create new mitochondria while also contributing to quality control by enabling the removal of damaged mitochondria and promoting apoptosis during high levels of cellular stress (Youle & Van der Bliek, 2012).

Mitochondrial biogenesis is mainly regulated by several key transcription factors, such as peroxisome proliferator-activated receptor gamma coactivator 1-alpha (PGC-1α), which regulates the generation of mitochondrial proteins. Mitochondrial dynamics are regulated mainly by a family of dynamin-like GTPases, of which dynamin-related protein 1 (Drp1) is one member. Drp1 is mostly localized to the cytoplasm but can translocate to mitochondria, where it participates in the division of the outer mitochondrial membrane (Losón et al., 2013). Two other members of this family are mitofusins (Mfn 1 and Mfn2), and optic atrophy type-1 (OPA1). Mfn1 and Mfn2 control the outer mitochondrial membrane, while OPA1 is located in and controls the inner membrane (Nan et al., 2017).

Normally, to maintain energy balance, mitochondria use a mechanism of adaptation to the metabolic needs of the cell - division and fusion. This process is still poorly understood; however its relationship with mitochondrial bioenergy is undeniable. Mitochondrial transcription factor A (TFAM), one of the critical regulators of mtDNA transcription, is associated with changes in mitochondrial fusion and mtDNA replication (Pohjoismäki et al., 2006). Thus, in patients with type 2 diabetes, an increase in the level of transcription of the TFAM gene and a decrease in its protein level in vitro models were noted (Pohjoismäki et al., 2006).

Balanced control of fusion and division is important for mitochondrial bioenergy (Gerald, 2019). Mitochondrial fusion processes are induced when optimization of mitochondrial bioenergy is needed. The fission processes are associated with the degradation of mitochondria. So fission is induced when mitochondria are damaged. Deficiency of proteins involved in mitochondrial fusion (Mfn2) reduces cellular respiration. Repression of Mfn2 is associated with a decrease in cellular metabolism, and a deficiency of the proteins, which involved in mitochondrial fusion, may indicate a decrease in cellular respiration and metabolism. Mitochondrial fusion may play a role in the development of oxidative stress in metabolic syndrome. Mitochondrial fusion mechanism rather than the fission mechanism was found is important for autophagy caused by glucose starvation. The lack of fission due to suppression of the expression of Drp1 leads to a loss of mtDNA and a decrease mitochondrial respiration in the cells. However, another study has demonstrated that inhibition of Drp1 prevents a decrease in mitochondrial membrane potential and the release of cytochrome C in cells. All these reports indicate that a shift to fission processes is associated with mitochondrial dysfunction in the main tissues, such as skeletal muscle and liver, involved in metabolic diseases associated with obesity (Wu et al., 2020). Several studies have shown that a change in the processes of mitochondrial fusion and fission can affect the production of ROS in the mitochondria. The balance between mitochondrial fusion and fission affects the important role in the regulation of mitochondrial energy and potentially contributes to maintaining normoglycemia in obese patients.

Thus, the balance between fusion and fission of mitochondria plays a vital role in the regulation of mitochondrial energy (Hu et al., 2020).

Factors governing mtDNA copy number in insulin resistance

Mitochondria respond to various metabolic disorders, such as oxidative stress, inflammation, glucotoxicity and lipotoxicity, biogenesis, and mitophagy, and the dynamics of mitochondria change (fusion and fission) (Koliaki & Roden, 2016). Scientific evidence indicates the existence of a compensatory mechanism, which manifests as an increase in the number of copies of the mitochondrial genome under the influence of pro-inflammatory factors (Xie et al., 2015). In particular, upon activation of TNF signalling pathway, the process of mitochondrial fission is triggered by DRP1 (Xie et al., 2015; Zhang et al., 2013; KEGG PATHWAY, 2019). Studies in animal models of obesity have shown that with prolonged exposure to ROS, mitochondrial biogenesis changes. Perhaps that in the first months of life, the number of mtDNA copies of increases to compensate for oxidative stress. But over time, intracellular stocks are being depleted. In this regard, the number of mtDNA was significantly reduced (Wang et al., 2014).

A mutation of a gene in a single mtDNA molecule does not affect the nucleotide sequence of the same gene in any other mtDNA molecule (Sivitz & Yorek, 2010). It is assumed that the dysfunction of mtDNA is not associated with damage to single copies of mtDNA but is associated with changes in the copies of mtDNA in the cell (Abdullaev, Antipova & Gaziev, 2009; Marín-García, 2016). In this regard, multiple copies of mtDNA may protect mitochondria from the severe damage to molecules under conditions of increased oxidative stress in the proximity of the ETC.

A mitochondrion is estimated to contain 2–10 mtDNA copies (Duan, Tu & Lu, 2018; Avital et al., 2012), and the number of mitochondria in a cell varies depending on the needs of the tissue or organ. mtDNA has a higher mutation frequency than nuclear DNA (Duan, Tu & Lu, 2018; Avital et al., 2012). Normal mtDNA molecules and their mutated counterparts often display a ‘co-existence’ situation, termed heteroplasmy (Duan, Tu & Lu, 2018; Avital et al., 2012). Heteroplasmic patterns change in different tissues and are inherent to the original nature of the tissue. The percentage of heteroplasmy correlates with the penetrance of disease phenotypes.

In our previous studies, it was shown that an increase in the number of mtDNA molecules in adipose tissue at different locations [mesentery (Mes), subcutaneous adipose tissue (SAT) and greater omentum (GO)] is associated with an increase in pro-inflammatory cytokines TNF- α Interleukin-6 (IL-6) and Interleukin-8 (IL-8) in the blood plasma of patients with obesity (Litvinova et al., 2019). Other authors obtained data on a small but significant decrease in the number of mtDNA molecules corresponding with an increase in Body Mass Index (BMI) and age. In addition, the development of type 2 diabetes in patients with obesity did not depend on the number of mtDNA molecules (Kaaman et al., 2007). However, Xu et al., (2012) showed that the amount of mtDNA negatively correlated with age, BMI, insulin level, homeostasis model assessment of insulin resistance (HOMA-IR) index, and cholesterol and triglyceride levels in patients with type 2 diabetes. These conflicting data could be due to the small number and heterogeneity of the studied groups. However, they do confirm the involvement of mtDNA replication in the pathogenesis of obesity and type 2 diabetes.

We identified the dynamics of changes in mtDNA copy number with BMI. The number of mtDNA copy numbers in visceral adipose tissue (Mes and GO) increased in patients with BMI > 35 kg/m2 relative to persons with a BMI of 30–34.9 kg/m2 (Skuratovskaia et al., 2019). However, in all patients with a BMI > 40 kg/m2, the number of mtDNA copies in the Mes was reduced compared with the number in the Mes of those with a BMI of 35–40.9 kg/m2 and was comparable to that of the control group. The decrease in mtDNA copy number found in our study could be considered a mechanism for adapting to the “soft disconnection” of the ETC, which also protects against the vicious cycle of ROS production. In addition, we identified higher levels of TNF- α in plasma and more significant mtDNA copy number in different tissues of patients with type 2 diabetes relative to levels and numbers in patients with obesity but not type 2 diabetes (Skuratovskaia et al., 2019).

This result is logical because antioxidant protective enzymes in tissues of different locations function with different levels of efficacy (Marín-García, 2016), and tissues in people with obesity may exhibit different reactions to genotoxic agents than their counterparts without obesity. In mice, antioxidant protective enzymes in the pancreatic islets are expressed at a lower level than in other cell types (Marín-García, 2016). In this regard, it is necessary to take into account the heterogeneity of responses by different tissues to cytotoxic damage. However, animal experiments have shown that changes in mtDNA copy number in peripheral blood leukocytes undergo processes similar to those in muscle tissue and hepatocytes, a finding that was also confirmed in our studies (Skuratovskaia et al., 2019).

Perhaps the number of human of mtDNA molecules in human blood is an important indicator of various metabolic disorders. Therefore, quantifying mtDNA in various biological samples can be used for predicting and evaluating the effectiveness of IR treatment.

The role of pro-inflammatory factors in the regulation of autophagy and mitochondrial fission and fusion

Fusion and fission repair and regulate mitochondrial damage (Geto et al., 2020). Fusion enriches damaged mitochondria with normal genome and proteins. This helps to avoid damage to the mitochondria. Fission, in turn, contributes to the destruction of damaged mitochondria by mitophagy (Westermann , 2012).

The violation of mitochondrial quality control to remove damaged and dysfunctional mitochondria leads to the accumulation of damage associated with membrane patterns (DAMPs) released from injured cells, cell-free mtDNA, N-formyl peptides and Cardiolipin (Zhang et al., 2010). Mitochondrial DAMPs can bind and activate membrane or cytosolic pathogen recognition receptors (PRRs) such as nod like receptors (NLRs), toll-like receptors (TLR), like those recognized by pathogen-associated molecular pattern (PAMPs) (Collins et al., 2004). Thus, it activates different early-phase inflammatory mediators like TNF-α, interleukins, interferon-gamma (IFN-γ) and ROS/RNS (Picca et al., 2017). In particular, TNF-α, secreted by adipose tissue, hepatocytes and Kupffer cells, contributes significantly to mitochondrial dysfunction, contributing to the production of ROS and RNS by inducible nitric oxide synthase. Wang et al. (2012) reported that RIPK1/3 activates mitochondrial phosphoglycerate mutase/protein phosphatase–PGAM5, which dephosphorylates Drp1 at ser637 and promotes Drp1 fission activity (Wang et al., 2012; Luo et al., 2017)—as a result, triggering TNF-α-induced necrosis.

Thus, in turn, it will have a deleterious effect of damaging mitochondria. The net result is leading to the depletion of ATP production and promoting a switch to anaerobic glycolysis (Morris & Berk, 2015). Therefore a combination of mitochondrial dysfunction plus upregulation of fission of mitochondria that produce ROS/RNS to an extent that exceeds antioxidant capacity, is lakely to be an initiating factors in inflammation, ageing, and age-related diseases (Hernández-Aguilera et al., 2013).

In addition to fission and fusion, the process of mitophagy plays an important role in maintaining mitochondrial biogenesis. Mitophagy is a selective form of autophagy. Mitophagy helps to remove damaged or dysfunctional mitochondria through lysosomal degradation. Studies have demonstrated that loss of autophagy/mitophagy can lead to a build-up of cytosolic ROS and mtDNA. That, in turn, can activate immune signalling pathways that ultimately lead to the releases of inflammatory cytokines, including Interleukin-1α (IL-1α), Interleukin-1β (IL-1β), Interleukin-18 (IL-18), type I IFN and macrophage migration inhibitory factor (MIF) (Harris et al., 2018). Moreover, the release of these cytokines can subsequently promote the release of others, including Interleukin-23 (IL-23) and Interleukin-17 (IL-17) (Harris et al., 2018).

An important process in maintaining normal mitochondrial function is the balance of auto-mitophagy in the cell. For most differentiated tissues, proper control of the content, distribution, and activity of mitochondria is the key to maintaining normal cellular functioning. Because dysfunctional mitochondria contribute to the development of IR, autophagy is essential to maintain the normal functioning of mitochondria and glucose metabolism. It has been shown that, against the background of IR, skeletal muscle and liver cells suppress the autophagy of organelles (Cho, Choi & Cho, 2017).

Mitochondria are central to apoptosis. Numerous studies have confirmed changes in the activity of apoptotic factors in different tissues under normal and pathological conditions. Activation of caspases, the main trigger of apoptosis, is controlled by Bcl-2 family proteins, which regulate the release of caspase activators from mitochondria. There are two classes of proteins in the Bcl-2 family: anti-apoptotic proteins (Bcl-XL, Bcl-w, Mcl-1, A1, Bcl-Rambo, Bcl-L10, and Bcl-G) and pro-apoptotic proteins (Bax, Bak, and Bok) (Estaquier et al., 2012; Parrish, Freel & Kornbluth, 2013; Cao et al., 2016). [51;52;53].

It has established that these processes during oxidative stress are characterized by Bax activation and the initiation of apoptosis (Petrasek et al., 2013). Excessive ROS can also cause oxidative damage to mtDNA, proteins, and phospholipids and induce apoptosis (Cao et al., 2016). For these reasons, there will likely be significant differences in cell type and the typical pathways that regulate the structure, function, and content of mitochondria and the fate of cells exhibiting increased mitochondrial autophagy. These factors complicate the study of mitophagy regulation, especially in pathological conditions.

Mitochondrial peptides

The change in mtDNA copy number and the induction of fission and fusion depend on the energy needs of the cell. Mitochondria can serve as a matrix for the synthesis of mitochondrial peptides. It is known that 13 proteins, all components of the ETC, are encoded by mtDNA. However, recent studies have shown that mtDNA contains previously unknown short open reading frames (ORFs) that extend the genetic diversity of mitochondria (Lee, Kim & Cohen, 2016). As a rule, ORF-encoded polypeptides of the nuclear genome possess significant biological activity, which can also be characteristic of mtDNA-encoded proteins.

The role of mitochondria as functional organelles and the signal molecules they produce is critical for cellular energy homeostasis, and mitochondrial dysfunction contributes to the pathogenesis of metabolic disorders. One of these molecules is protein humanin (Fig. 2) (Sreekumar et al., 2016), which contributes to cell protection during oxidative stress by regulating mitochondrial functioning for more efficient ATP synthesis (Sreekumar et al., 2016). Increased mitochondrial biogenesis (in particular, the formation of new mitochondria, an increase in mtDNA copy number and the expression levels of mitochondrial transcription factors) in various insulin-dependent tissues may be one possible mechanism for optimizing bioenergy in cells under the action of humanin (Sreekumar et al., 2016). In contrast, humanin inhibits increases in mtDNA copy number in blood cells (Fig. 2) (Kim et al., 2017).

Figure 2 Diagram of the effects of mitochondrial peptides MOTS-c and humanin.

The peptide has positive effects in the regulation of carbohydrate metabolism and can interact with the nuclear genome. nDNA –nuclear DNA, mtDNA –mitochondrial DNA. This figure has been created by modifying the templates from Servier Medical Art (Servier Medical Art, 2020); CC BY 3.0 Unported.

We previously showed that, in patients with a high BMI, an increase in circulating TNF- α contributes to the activation of a compensatory mechanism for maintaining mitochondrial biogenesis. This compensatory mechanism is associated with increased mtDNA copy number in the cells of subcutaneous adipose tissue and reduced mtDNA copy number in blood mononuclear cells (Skuratovskaia et al., 2019). In this regard, mitochondrial peptides play an important role in the functioning and biogenesis of mitochondria.

A relatively recently identified peptide is MOTS-c (mitochondrial open reading frame of the 12S rRNA-c), which encodes a peptide of 16 amino acids. MOTS-c transcription occurs in the cytoplasm, and the mechanism by which polyadenylated MOTS transcripts are exported out of mitochondria is currently being investigated (Fig. 2) (Lee, Kim & Cohen, 2016). Although the functional activity of MOTS-c has not been extensively studied, several studies have examined the mechanism of action of MOTS-c in cell cultures and animal models; however, to date, its properties have been extensively studied in humans. MOTS-c affects mitochondrial metabolism and insulin sensitivity (Lee, Kim & Cohen, 2016), accelerates glucose uptake (Lee, Kim & Cohen, 2016; Lee et al., 2015), and activates insulin-dependent AKT kinase (Kim et al., 2017) in mouse skeletal muscle cells in culture. Also MOTS-c affects glucose production in the liver (Kim et al., 2017; Li et al., 2015).

One study showed that the level of MOTS-c in the circulation was reduced in male children with obesity and was associated with markers of IR and obesity (Childress et al., 2018). Another research group measured plasma MOTS-c concentrations in healthy individuals and patients with obesity, and although they found that the protein level did not differ, they also found a negative correlation between plasma MOTS-c and the HOMA-IR index and the Matsuda index (alternative index of insulin sensitivity) (Cataldo et al., 2018). However, the samples in these studies were limited to a small number of patients.

Under the influence of cellular stress, MOTS-c can translocate to the nucleus, where it affects the nuclear genome by changing its expression, which is an action particularly uncharacteristic of proteins encoded by mtDNA (Mangalhara & Shadel, 2018). In the nucleus, MOTS-c binds to DNA and regulates gene transcription in combination with other transcription factors that contribute to protection against stress. This mitochondrial peptide is the first to have been discovered acting on transcriptional reactions in the nucleus in response to stress. MOTS-c regulates the expression of nuclear genes during metabolic stress in a 5′-adenosine monophosphate protein kinase (AMPK)-dependent manner (Fig. 2) (Kim et al., 2018). Therefore, MOTS-c may have some characteristics of a “mitokine” factor (Fig. 2) (Yong & Tang, 2018; Mendelsohn & Larrick, 2018).

MOTS-c inhibits inflammatory pathways and improves the functioning of the endothelium (Li et al., 2018). Although the relationship between MOTS-c and the activation of the NF-kB receptor (Li et al., 2018; Ming et al., 2016) has been shown, the nature of this relationship has not yet been defined. MOTS-c inhibits the folate cycle involved in the regulation of carbohydrate metabolism by reducing the biosynthesis of purines, which leads to accumulation of the intermediate product of purine synthesis, 5-amino-imidazole-4-carboxamide (AICAR), and activation of the metabolic regulator 5′-AMP-activated protein kinase (AMPK) (Fig. 2) (Lee et al., 2015). Folate serves as a source of single carbon units for the methionine/homocysteine cycle by supplying 5-methyltetrahydrofolate used for the methylation of homocysteine such that it is reverted to methionine (Zhao et al., 2018). For example, decreased methionine can induce hepatic lipid accumulation by downregulating sterol regulatory element-binding protein (Srebp1) mRNA and upregulating the expression of acetyl-CoA carboxylase 1 (Acc1) and fatty acid synthase (Fasn) mRNA, which are involved lipid synthesis in a hepatic cell (Zhao et al., 2018). The main risk factors for insulin resistance are hepatic steatosis. Moreover, insulin resistance can lead to the development of type 2 diabetes (Zhao et al., 2018). Thus, it was speculated that folate deficiency could induce glucose and lipid metabolism disorders.

It has been shown that MOTS-c works by regulating signalling pathways that act through AMPK and Sirtuin 1 (SIRT1). It should be noted that at high concentrations of MOTS-c, glucose is incorporated into the pentose phosphate pathway, which ensures the production of the substrate for the synthesis of purines, and is not oxidized through glycolysis (Lee et al., 2015). With this in mind, the use of MOTS-c for the correction of carbohydrate metabolism is of great interest.

In addition, MOTS-c may serve as a potential therapeutic agent in the treatment of sepsis. Sepsis is characterized by uncontrolled inflammatory reactions to pathogenic bacterial infections, especially antibiotic-resistant strains. A study showed that MOTS-c increased survival rate and decreased bacterial load in infected mice, a decrease in the pro-inflammatory cytokines TNF-α, IL-6, and IL-1β and an increase in the anti-inflammatory cytokine Interleukin-10 (IL-10) (Zhai et al., 2017).

Mitochondria are directly involved in changes in carbohydrate metabolism and the development of IR. Given the promising positive effects of MOTS-c on the regulation of metabolic homeostasis, the therapeutic effects of this peptide in relation to obesity and diabetes are obvious. Future studies should focus on studying the metabolism and activity of MOTS-c in insulin-sensitive tissues in patients with type 2 diabetes mellitus to develop diagnostic methods and determine the therapeutic potential of this peptide.

Therapeutic strategies based on mitochondrial regulation

Mitoproteomics in the treatment of insulin resistance

The development of methods for proteomic analysis has allowed the investigation of changes to the composition of mitochondrial proteins, their redox state, and how the proteins interact with each other and regulate the functions and dynamics of mitochondria.

In mammalian cells, mtDNA encodes 13 proteins belonging to the mitochondrial respiratory chain, 2 ribosomal RNAs, and 22 transport RNAs, which determines the intramitochondrial translation code. This pathway is strictly regulated to maintain mitochondrial function (Gustafsson, Falkenberg & Larsson, 2016).

The remaining mitochondrial proteins, which make up the other 99%, are encoded by nuclear genes and depend on specific signals that direct them from the cytosol, where they are synthesized, to receptors on the surface of mitochondria and then to the corresponding mitochondrial compartment (Pfanner, Warscheid & Wiedemann, 2019). These include proteins involved in β-oxidation and protein transport; apoptotic factors; respiratory chain subunits, and the TCA components (Gómez-Serrano et al., 2018).

There are mechanisms by which mitochondrial proteins are imported into the organelle (Chinnery & Hudson, 2013; Chacinska et al., 2009; Stojanovski, Bragoszewski & Chacinska, 2012). The import of cytosolic-synthesized mitochondrial proteins usually requires cytosolic chaperones, such as heat-shock proteins (HSP) 70 and 90. These chaperones direct pre-proteins to the receptors of the translocase in the outer mitochondrial membrane (TOM) complex, mainly TOM70 and TOM20 (Neupert & Herrmann, 2007).

Using a new approach, redox mitoproteomics, the authors proved that disruption of the mitochondrial complex assembly, together with the import of defective proteins, can be a potential cause of mitochondrial dysfunction in type 2 diabetes (Gómez-Serrano et al., 2017); for example, changes in the pathways involved in the import of mitochondrial proteins at different levels, such as the TOM complex or the path of the mitochondrial intermembrane space assembly (MIA) (Gómez-Serrano et al., 2017; Stojanovski et al., 2008), may be involved. Also the authors established that complex IV (CIV) is a common target for the mitochondrial remodelling induced by ageing or type 2 diabetes (Gómez-Serrano et al., 2017).

The study of proteomic profiles provides unique information about thousands of mitochondrial proteins that can be used to assess mitochondrial functions that are related to the development of type 2 diabetes (Chae et al., 2018). The large size of the proteome indicates that many metabolic and cellular pathways are active in mitochondria. Recent studies have shown significant tissue-specific differences in the composition of mitochondrial proteins (Mootha et al., 2003; Pagliarini et al., 2008). Approximately 1/3 of all mitochondrial proteins are key components of the OXPHOS subunits and the TCA. Most subunits are tissue-specific.

In patients with obesity, studies of the mitoproteome within the fatty tissue of the greater omentum have shown that mitochondrial protein levels are reduced compared to those of the controls (without obesity). In addition, four proteins [citrate synthase, mitofilin, hydroxyacyl-CoA dehydrogenase/3-ketoacyl-CoA thiolase/enoyl-CoA hydratase (HADHA), and mitochondrial proton/calcium exchanger protein (LETM1)] were inversely correlated with BMI (Chae et al., 2018; Lindinger et al., 2015). Chae et al. (2018) investigated the mitochondrial proteome of skeletal muscles in patients with type 2 diabetes and compared the 1150 proteins discovered with the proteomes of mice and humans published elsewhere. It was established that 592 of the proteins had not yet been described, indicating that a wide range of mitoproteomic insulin-dependent tissues is involved in various pathophysiological processes.

A comparison of mitochondrial proteomes in patients with type 2 diabetes and those without diabetes revealed 335 differentially expressed proteins. Of these, 135 proteins were different in type 2 diabetes compared to those of the control (without type 2 diabetes), and 200 proteins were less active. Based on the results of extensive work in this area, the authors identified five potential proteins that indicate dysregulation of mitochondrial functions in type 2 diabetes: three proteins are associated with enhanced functions of the mitochondria-associated ER membrane (MAM), SORT, CALR, and RAB1A; and two proteins are associated with OXPHOS, mitochondrial NADH dehydrogenase ubiquinone iron-sulfur protein 3 (NDUFS3) and cyclooxygenase (COX2).

Thus, a comprehensive study of mitochondrial proteomes in insulin-sensitive tissues can provide a molecular basis for understanding the metabolic and cellular pathways that link mitochondrial dysregulation with type 2 diabetes.

Therapy for regulating mitochondrial function

Drugs and drug transport vehicles like aimed at mitochondria have significant potential because of the presence of molecular target redundancy and their key role in metabolism (Murphy & Hartley, 2018). Mitochondrial dysfunction can lead to pathologies that affect several pathways: ATP supply, mitochondrial biogenesis, mitochondrial fission and/or fusion, and it can affect the quality control of organelles. Drugs can directly affect the mitochondria or indirectly affect mitochondrial function by binding to regulatory targets in the cytosol or nucleus (Nightingale et al., 2016; Smith et al., 2012).

Metformin is an oral hypoglycaemic biguanide drug currently used to treat type 2 diabetes. It inhibits mitochondrial complex I of the ETC and is an activator of AMP-activated protein kinase; therefore, metformin can reduce ATP production (Fig. 3) (Hawley et al., 2002).

Figure 3 Target drags for the regulation of biogenesis and functioning of mitochondria.

The effects of drugs regulating mitochondrial function. This figure has been created by modifying the templates from Servier Medical Art (Servier Medical Art, 2020); CC BY 3.0 Unported.

Low molecular weight mitochondrial uncouplers act as protonophores, transporting protons into the matrix independently of protein complexes or mediating molecular dissociation using proteins such as adenine nucleotide translocase (ANT) (Fig. 3) (Childress et al., 2018). The use of low molecular weight uncouplers, such as dinitrophenol (DNP), which reduces the proton motive force (Δp) in the inner mitochondrial membrane, makes OXPHOS less efficient, which helps burn excess fat tissue and reduces the mitochondrial ROS levels (Harper et al., 2001).

However, using pharmacological agents to uncoupled all mitochondria throughout the body may be a high-risk treatment. This is due to a might compromise energy homeostasis in tissues, especially the heart and brain. In humans treated orally and in animals, it was shown that the drug promoted a direct stimulation of cellular respiration and a consequent rise in body temperature, and also a significant decrease in body weight due to loss of body fat (Magne, Mayer & Plantefol, 1932). Therefore uncoupling can increase energy expenditure without compensatory mechanisms. Thus, the main shortcomings of the drug are a small difference between the effective and fatal doses of DNP, as well as a nonselective actions (Harper et al., 2001).

The therapeutic potential of mitochondrial uncouplers is associated with their dual role in increasing the oxidation of nutrients and decreasing the amount of ROS produced through the ETC. Although increased oxidation of nutrients contributes to weight loss and is a therapeutic strategy for the treatment of obesity and related metabolic diseases (Perry et al., 2013), mitochondrial ROS are associated with numerous pathologies, including ischaemia, reperfusion injury, inflammation, insulin resistance, neurodegenerative diseases, and many other pathologies. It is important to note that mitochondrial uncouplers prevent the production of ROS, which is beneficial, compared with antioxidants that remove ROS that has already been produced (Fig. 3). Therefore, a decrease in the production of mitochondrial ROS has significant therapeutic potential and advantages over antioxidant acceptors.

The balance of ROS and ATP depends on the rate of reduction of one electron O2 to a superoxide O2. − through the ETC and on the absorption rate through antioxidant systems. Mitochondria maximize energy production when ROS concentrations are high to create an intermediate redox state. Over time, the balance is lost, and ROS overflow occurs, which increases with distance from the optimal redox potential. At mitochondrial redox potentials rarely found, ROS production exceeds the absorption capacity such that, under oxidative conditions (for example, at high loads), the antioxidant protection is compromised and ultimately eliminated (Aon, Cortassa & O’Rourke, 2010).

The study of mitochondrial dynamics was proposed as a means to find the link between mitochondrial dysfunction and IR (Lin et al., 2018). Overexpressing proteins associated with fusion and inhibiting proteins associated with fission strengthens the network of mitochondria and suppresses IR, activating the IRS1-Akt pathway and inducing GLUT1/GLUT4 translocation to the cell membrane. Pharmacological inhibition of DRP1 using the mitochondrial division inhibitor (Mdivi-1) reduces IR and could initiate a new era in diabetes treatment (Lin et al., 2018). Mdivi-1, a chemical compound that weakens mitochondrial fission by selectively blocking the GTPase activity of DRP1 (Cassidy-Stone et al., 2008), eliminates the dynamic imbalance of the mitochondrial genotype and weakens insulin signalling in insulin-dependent tissues. A significant treatment Mdivi-1 restores the balance of mitochondrial fusion and division in sepsis. The underlying molecular mechanism may be that Mdivi-1 prevents the self-assembly of Drp1, which is required for mitochondrial fission, and the selective inhibition of mitochondrial fission, in turn, suppresses mitochondrial fusion defects (Tanaka & Youle, 2008; Pagliuso, Cossart & Stavru, 2018). Also, Mdivi-1 reduced the production of ROS and prevented the occurrence of the endoplasmic reticulum stress in inflammation.

Mitochondrial biogenesis can be enhanced using drugs that indirectly alter the activity of PGC1α (Whitaker et al., 2016; Scarpulla, 2011). AMPK agonists, such as AICAR, activate PGC1α, mimicking enhanced mitochondrial biogenesis (Viscomi et al., 2011). Another approach is the use of the SIRT1 activators resveratrol and viniferin, which activate PGC1α by reversing acetylation (Whitaker et al., 2016). A parallel approach to enhancing mitochondrial biogenesis is to inhibit pathways that suppress it, such as that of the hypoxia-induced factor 1 α (HIF1α) (Zhang et al., 2007; Semenza, 2007).

The main problem in the development of drugs aimed at mitochondria is the need to create a suitable way to deliver the desired molecule to its intended target. The development of drug delivery systems that can permeate the mitochondrial double membrane is the key to the success of mitochondrial therapy. Small molecules, such as ions, ATP, and proteins smaller than 5 kDa, can freely diffuse through the channels of the outer membrane (proteins smaller than 10 kDa through pass the TOM import gate); however, the mitochondrial inner membrane forms a barrier that selectively transports molecules into the mitochondrial matrix (Jang & Lim, 2018).

There are delivery methods that use delocalized lipophilic cations. Such cations are connected to the negatively charged mitochondrial matrix and can cross the membrane for accumulation in the mitochondria (Murphy, 1997). Peptides containing the mitochondrial directional sequence, which recognizes receptors on the mitochondrial membrane, have been used successfully to transport chemical cargoes into the mitochondria (Vestweber & Schatz, 1989). Viruses are widely used in gene therapy as carriers because of their innate ability to insert their genome into a host cell. For example, the vector of adeno-associated viruses has been used to deliver genes to mitochondria (Yu et al., 2012); however, despite numerous methods, few are sufficiently effective, and each has drawbacks.

In recent decades, the emergence of a completely new field of mitochondrial medicine is based on an acknowledgement of the important role that mitochondria play in both human health and disease. Molecules for the development of these treatments must selectively act and accumulate on the target site.

Therefore, it is necessary to investigate mitochondria as a target for treatment and to understand the pathophysiology of the disease. Treatment strategies aimed at enhancing mitochondrial function may represent important new approaches in the treatment of diabetes. The discovery of the main mechanisms involved in the pathogenesis of diabetic complications provides a new conceptual basis for future research. However, clinical trials will be required to show that the compounds used in cell culture and animal studies apply to humans.

Conclusions

Mitochondria play an important role in the regulation of energy metabolism, ROS production, apoptosis, and signal transduction. Mitochondria are dynamic organelles and can reprogram themselves depending on the needs of the organism, regulating mtDNA stability, respiratory function, apoptosis, cell stress reaction, and mitochondrial degradation. The dynamic process of mitochondria may not be balanced as a result of fusion and fission proteins. This leads to the accumulation of damaged mitochondria producing ROS. ROS, produced by dysfunctional mitochondria, damages the mitochondria of various organs and tissues that cannot function properly, which leads to chronic and age-related disorders.

Analysing the behaviour of factors that have systemic effects on interdependent mechanisms and cascades of intracellular signalling pathways can broaden the fundamental understanding of the components of the signalling network. In addition, the pathogenetic mechanisms of diseases such as diabetes and degenerative diseases characterized by mitochondrial dysfunction can be studied from a new perspective. The final clarification of the mechanisms of energetic homeostasis will enable the use of effective individually selected therapies based on the physiological characteristics of insulin-sensitive metabolism.

Additional Information and Declarations

Competing Interests

Author Contributions

Data Availability

The authors declare there are no competing interests.

Daria Skuratovskaia conceived and designed the experiments, analyzed the data, prepared figures and/or tables, authored or reviewed drafts of the paper, and approved the final draft.

Alexandra Komar and Maria Vulf performed the experiments, prepared figures and/or tables, and approved the final draft.

Larisa Litvinova performed the experiments, analyzed the data, authored or reviewed drafts of the paper, and approved the final draft.

The following information was supplied regarding data availability:

We provide a literature review without quantitative data.

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
