# Peer review of "Mitochondrial destiny in type 2 diabetes: the effects of oxidative stress on the dynamics and biogenesis of mitochondria"

_PeerJ, doi:10.7717/peerj.9741_

## Round 0.1 · original submission · Major Revisions

As you will see from the reviews, both experts in this field have found your review to be interesting and worthwhile. Both they and I agree this deserves to be published as it makes an attempt to synthesise many different aspects of mitochondrial biology under the theme of diabetes. However, we all tend to think the review struggles a little with focus as a consequence, in part, of its breadth.

Hence, I have returned this to you as 'major revisions' as I think there is a good deal of editorial work required to tidy this up and aim for greater brevity. I do not think this will present you with great difficulty, but as it does need extensive changes I chose this category.

However, this should not detract from what we regard as a good review of an important subject which we are keen to publish. Hence, both the reviewers and I urge you to re-cast your review as suggested. I am sure it will then be viewed favourably.

Thanks for submitting this work to PeerJ.

·

Basic reporting

This manuscript is generally well-written and clear throughout – with occasional specific sections that have been identified in General Comments. The background to the subject is clear and literature well referenced and relevant. The structure is sensible, although the manuscript would benefit from work to increase brevity. The review topic is cross-disciplinary and of excellent potential interest to a broad audience; there are multiple reviews published on various aspects of the manuscript, however very little that draws them together with the focus of the current manuscript. The Introduction adequately introduces the subject (although the description of mitochondrial OXPHOS lines 43-53 seems unnecessary – is textbook – and could be replaced by a brief introduction to insulin resistance/adipocytes).

Experimental design

I am not sure this manuscript is really a systematic review rather than a regular review with search terms included in methodology. The search does not seem reproducible by externals – 20 keywords with unspecified combinations and subjective eligibility criteria. page 7, lines 68-74. I don’t think that this undermines the manuscript – but rather just suggests a different label from “systematic review” to “review”. Literature included does appear to be unbiased and comprehensive, and is cited appropriately. Organisation is logical.

Validity of the findings

Argument is well developed. Conclusions are well stated and indentify future direction – although some specific queries are highlighted in “General comments” below.

Additional comments

The manuscript is fairly long and would be improved by careful condensation and brevity.
Major points:
• p8, line 93: “Signalling molecules can interact with effectors on the surface of mitochondria” is very vague and does not appear to be explained subsequently – other than an indication that oxidative/nitrosative stress can cause mitochondrial dysfunction and further ROS production. The sentence prior to line 93 discusses transcription factors activated during inflammation, and alterations in insulin signalling molecules. If the authors are implication mitochondrial involvement purely through ROS/RNS then this section should be made less ambiguous.
• p9, line 126 on: the stated dogma of fusion alleviating stress and fission contributing to quality control/mitophagy is now somewhat more nuanced, recent evidence has also implicated fusion (or at least Mfn2) in mitophagy, mito transport and interactions with ER (reviewed in Dorn 2019, https://doi.org/10.1146/annurev-physiol-020518-114358, and also see the very recent: doi: 10.3389/fcell.2020.00221 and doi: 10.1080/15548627.2020.1749490).
• p9, lines 139-150: these paragraphs more appropriate under sub-heading 1.2 as they don’t mention mito fission/fusion at all? Or maybe adapt this section by adding in discussion of literature that implicates mito dynamics/cristae morphology – and OPA1 in particular – in maintenance of mtDNA?
• p8, line 110 and p10, line 184: by “soft separation / disconnection of the ETC”, do you mean mild uncoupling of ETC that decreases correlation of metabolic input with ATP/ROS production?
• Figures 2 and 3 could be clearer by making arrows more distinct and re-arranging the diagrams where-ever possible to minimise arrow cross-over. Do make sure that the figure legends support the whole of the figure.
• Discussion of ref 59 on p13-14 could be made both briefer and clearer (in particular “135 proteins were different in type 2 diabetes compared to those of the control (without type 2 diabetes), and 200 proteins were less active” = “135 proteins were upregulated in type 2 diabetes compared to those of the control (without type 2 diabetes), and 200 proteins were downregulated” – I don’t think that the paper measured activities?)
• The majority of section 4 is quite basic knowledge – could the essential aspects be incorporated into a short intro paragraph of section 5 instead? (ie lines 350-351 and 366-8, with very concise supporting text?)
• The discussion of the potential benefits of uncouplers such as DNP in T2D (p17) does not include any mention of the well-known dangers of this drug (numerous deaths due to overheating and energy failure that accompany uncontrolled uncoupling when taken illicitly). Equally, description of Mdivi-1 should potentially be tempered by mention of this drug’s ability to induce apoptosis.
Minor points
• p8, line 81 “a result of IR or whether insulin resistance itself” – use abbreviation throughout once introduced
• p8, line 84 “hypertrophic increase” = “hypertrophy”?
• p25 Figure 1:
o legend should make reference to A and B to direct reader around figure;
o also explain DAMPS and PAMPS; also “violate insulin signalling” seems somewhat dramatic and uninformative – “block/prevent/impinge on/downregulate insulin signalling leading to a decrease in the intracellular pathways that stimulate GLUT4 translocation to the plasma membrane”?;
o mitofusins 1 and 2 are show in the figure as “Mtf-1” and “Mtf-2” however are referred to via the standard “Mfn-1/2” within the main text.
• p8, line 95: “which are is associated”
• p8, line 113-4: “… based on the effects of certain drugs used to treat type 2 diabetes. The drug mechanism of action is aimed at improving insulin sensitivity …” grammar not great, and meaning somewhat unclear; should the second sentence read “Among the mechanisms of action of these drugs that lead to improved insulin sensitivity, is the suppression of ATP generation …”?
• p9, line 137: “Two other members of this family are mitofusin 1 (Mfn1), and optic atrophy type-1 (OPA1). Mfn1 and Mfn2 control…” surely better as “Two other members of this family are mitofusins (Mfn1 and Mfn2), and optic atrophy type-1 …”?
• p11, line 230: “to date, its properties have been extensively studied in humans” – should this be “have not been”?
• p12, line 239: “correlation between the HOMA-IR index and the Matsuda index” clarify that this means correlation of MOTS-c with EACH, and also insert some explanation of Matsuda index – even just “the alternative insulin-sensitivity index Matsuda index”.
• p12, line 250: this statement is attributed to ref 46 – which is actually a commentary on the primary research paper Qin et al Int J Cardiol. 2018 Mar 1;254:23-27. doi: 10.1016/j.ijcard.2017.12.001, but this paper is not cited. Is there a reason for this? (I cannot easily access ref #46 just now)
• p13, line 296-7: redundant as covered in paragraph above.
• p13, line 304 & 306: “importation” = import?
• p13, line 318 does not make grammatical sense
• p16, line 402: the sentence starting “Mitochondrial fusion….” is unnecessary as it just repeats information from earlier in the review. (indeed, this whole paragraph could be much tighter)
• p16, line 421: “Drug transporters aimed at mitochondria” is ambiguous – drugs that are transported in to target mitochondria? or mechanisms for transporting drugs in? or just drugs targeted at mitos?
• p16, line 434: “proton force (Δp)” – meant to be proton motive force?
• p18, line 487: “To targeting molecules involved…” this sentence does not make grammatical sense.

Reviewer 2 ·

Basic reporting

This review covers a wide range of issues linking mitochondrial function and type 2 diabetes. The authors have summarised a great deal of work and provided an interesting review article. The style and language is clear (although some of the content is presented in a fragmented manner see comments below). The background and literature is covered sufficiently, but sometimes lacks depth. The work is provided in good context. Figures are clear and well integrated with the text. There are several reviews on mitochondria biogenesis and assembly, but not many on the relevance of mitochondria function to T2D. I find the title a little vague (mitochondrial 'variations' may mean different thing to dfferent people and the scope of the review later on is heavy on mtDNA aspects.

Experimental design

Overall logical presentation of material albeit a little disjointed.
Specific points:
-lines 206-208: not clear the relationship between mtDNA and nDNA. A balance in the expression of the two genomes for mitochondrial protein is needed, but this is not discussed in any great detail.
-line 209: they authors mention 'signal molecules' and then go on to say 'one of those proteins': signal molecules generated by mitochondria can be distinct ROS, metabolites of the TCA cycle, proteins etc. It is not clear whether the authors will refer to all of those or not.
- lines 222-224: the generalisation is too far fetched, as the cases of mitochondrial derived peptides are only two mentioned here.
- the section 5 on chemerin is introduced too abruptly in the text and seems out of place.
- line 421: 'drug transporters'??? Reading the rest of this section I understood that the authors mean modulations or drug transport vehicles like the TPP-moiety etc but not transporter proteins. This is confusing.
- line 472: the statement that proteins smaller than 10 kDa can freely diffuse across the OM is not correct. Small molecules less than 5 kDa are freely diffusible through the VDAC channels but not proteins smaller than 10 kDa. In fact there are several examples of proteins around 10 kDa (eg many substrates of the MIA pathway) that do not freely diffuse but through the TOM import gate
- line 475: the inner membrane 'consists of cardiolipin' is inaccurate: cardiolipin is in high concentrations in the IMM, but it is also present in the plasma membrane, and the IMM also contains other lipids.
- lines 475-488: it should be clearer that this whole discussion here concerns the targeting not to mitochondria in general but to the specific matrix sub-compartment. In fact none of the technologies mentioned can target any molecule to the intermembrane space or indeed to the OMM or the IMM.
- reference 91: historically, the capacity for DNA to be delivered to the mitochondrial matrix when fused to a matrix protein was shown already in 1989 (Vestweber and Schatz, 1989, Nature 338:170-2

Validity of the findings

Arguments are mostly well developed. Conclusions are valid, and relevant literature referenced.

Additional comments

This is a nice effort to bring together several aspects of mitochondrial biology and their relevance to T2D. I find the review interesting. It is hard to follow in parts because it tries to cover different aspects (mitochondrially derived peptides, dynamics, mito proteomics, autophagy). All of these could be in fact separate reviews in their own right. However, scanning all of these aspects has some value for people working in T2D as it highlights the multitude of ways mitochondria can be linked to this disease. To streamline the review I would suggest taking out the part on chemerin as it does not really fit so well, and takes the reader to yet another line of research (mitochondria and inflammation).

---

## Round 0.2 · Minor Revisions

As you will see, both reviewers are now happy to recommend acceptance, but there remain a few points of English and clarity which require attention. Could you kindly correct these and then re-submit?

·

Basic reporting

Good, no further comment specifically on basic reporting, see below for general comments.

Experimental design

Good, now has appropriate designation as review rather than systematic review.

Validity of the findings

Good, no further comment specifically on basic reporting, see below for general comments.

Additional comments

I appreciate the changes that the authors have made in line with reviewers’ comments. I think that the text is now more focussed and the supporting figures are much clearer for the reader.

General points:
- could do with overall check of grammar and flow of wording throughout, following authors' Track Changes, particularly focussed on newly-written or adapted sections - eg p2, line 43 "Process oxidative phosphorylation (OXPHOS) enables ATP generation in mitochondria. Membrane electrochemical gradient stimulates ... " should be "The process of oxidative phosphorylation (OXPHOS) enables ATP generation in mitochondria. A trans-membrane electrochemical gradient stimulates ...."

Minor points noticed:
-p4 line 156 (with “All Markup” selected) – mitofusins (Mfn and Mfn2) = mitofusins (Mfn1 and Mfn2)
-p5 line 172 – “Mitofusin may play a role” = “Mitofusins may play a role” or “Mitofusin-2 may play a role” or “Mitochondrial fusion may play a role”?
-p6 paragraph starting line 200 – check where underlining is intended
-p7 line 258 – “Perhaps that was the number of human mtDNA in blood cells is an important indicator of various metabolic disorders” does not make grammatical sense – “perhaps the number of mtDNA molecules in human blood ….”?
-p8 line 284 – not clear what this means: “Therefore, mitochondrial dysfunction with fission upregulation prevents the elimination of damaged mitochondria producing ROS and RNS that exceed the antioxidant activity is likely initiating factors in inflammation, aging, and age-related diseases [48].” Should it be something like “Therefore a combination of mitochondrial dysfunction plus upregulation of fission of mitochondria that produce ROS/RNS to an extent that exceeds antioxidant capacity, is likely to be an initiating factor in ….”?

Reviewer 2 ·

Basic reporting

The revised version has taken into consideration hte majority of the points raised. Shortening the review also helps in focus.Broad interest and good reviewing of the jcurrent knowledge in the foeld sufficiently covered.

Experimental design

Good citing of literautyre, clealry improved in the revision.

Validity of the findings

Clear logic, good flow of the text

Additional comments

I am pleased to see the revision and the points that have been addressed by the authors. This is a much improved version which I am happy to recommend for publication.

---

## Round 0.3 · accepted · Accept

Thanks for finishing up the final points.